# Mental Representations and Cognitive Schemata of Ninth Grade Students for the Refraction of Light

George Fyttas [1], Vassilis Komis [1], George Kaliampos [2] and Konstantinos Ravanis [1,*]

[1]   Department of Educational Science and Early Childhood Education, University of Patras, 26504 Patras, Greece; gfyttas@upatras.gr (G.F.); komis@upatras.gr (V.K.)

[2]   Department of Education, School of Education, University of Nicosia, 2417 Nicosia, Cyprus; kaliampos.g@unic.ac.cy

[*]   Correspondence: ravanis@upatras.gr

**Abstract:** The current research study deals with students' mental representations and cognitive schemata of light refraction. In the study, 213 ninth grade students participated who had taken basic Geometric Optics courses on refraction and Snell's law. The students were given three tasks in which they were asked to predict and explain the phenomenon of refraction. The results showed that the vast majority of them articulated their responses based on representations that were not compatible with the Geometric Optics model. Quite interestingly, the Multiple Correspondence Analysis led to five distinct cognitive schemata resulting from a fixed combination of representations.

**Keywords:** refraction; light; students' mental representations; students' cognitive schemata

## 1. Introduction

Research on students' understanding of the phenomena of optics over the past 30 years has led to the development of a distinct area of research in Science Education. Phenomena and concepts such as light as an entity, linear propagation, shadow formation, vision, image formation and reflection [1–5] constitute some of the most common research topics within this area. They are usually approached within the framework of the Geometric Optics model, which is based on the assumptions of a point light source and isotropically propagating light rays [6]. This model is dominant and is commonly used in the education of children 5–6 to 15–16 years old [7–9].

The transformation of scientific models into appropriate school models offers significant advantages in learning, with the ability to formulate stable reasoning being one of the most prominent among them. The syllogisms here are based on three important functions that become powerful tools of thought: description, interpretation and prediction [10,11]. The Geometric Optics model possesses without deviation these syllogisms and thus constitutes a very suitable framework for teaching and learning. However, there are phenomena that are incorporated into the curriculum of primary and secondary education, such as reflection and refraction, which explanation require the usage of the Wave Optics model, even if their descriptions and predictions can be made with the basic elements of Geometrical Optics. Given the fact that this model is usually introduced at the end of secondary education while reflection and refraction are studied much earlier, it is apparent that specific tools developed within the context of Geometrical Optics are exploited for teaching these phenomena. One of them is Snell's law, which gives us the capability to measure the deviation of a light ray when passing from one medium into another through establishing a relationship between the path taken by the light ray in crossing the surface of separation between two contacting substances and the index of refraction.

It is well known in a Science Education research context that students approach the phenomena of the natural world and the concepts of Physical Sciences through the formation of mental representations that are not compatible with scientific knowledge and

are resilient to both biological maturing and traditional teaching. Despite the importance for teaching of recording and classifying mental representations, they are naive, implicit, local and non-conscious [12,13]. However, in spite of their weak nature, representations are often combined in a way that allow types of reasonings that are characterized by stability, since they are used in a number of different situations, regardless of whether they are compatible with the school–scientific model or not. The stability of these forms of reasoning leads to the recognition of them not just as simple representations but as cognitive schemata, i.e., cognitive organized units of knowledge that operate as a framework for the approach of subjects, objects, situations, events and phenomena [14–16]. The term 'mental representation' was chosen among others such as 'alternative idea', 'misconception', etc. This schematization of naïve entities of children's thinking signifies not only a specific knowledge content but an entity included into a cognitive system with concrete structures that are 'processing and mapping' procedures. In that kind of system, the term 'mental representations' is considered more suitable, since it approaches the structural parameters, as well as the functional interrelations of a wider system [17]. In the same perspective, the concept of 'cognitive schema' was chosen. Based on Piaget's [18] approach, the cognitive schema of a mental operation is its general structure—that is, what is kept unchanged in repetitions of the same operation, adapted to different situations and stabilized with practice. The cognitive scheme consists of organized elements of different past experiences and situations that form a relatively cohesive and persistent frame of knowledge that guides one's subsequent perception and consideration of the world [19].

Apparently, the relation of representations to schemata is not a given generalized possibility but a potential conquest for the development of thinking in some specialized fields that need systemic investigation. Along this line, in the current article, we attempt to study the transition from mental representations to schemata for the phenomenon of light refraction. Research on how students up to 15 years of age deal with refraction highlights the existence of major difficulties in the conceptualization of this phenomenon.

Anderson & Kärrqvist [20] studied the representations of 12–15-year-old students about light by posing the task of observing an object at the bottom of a container full of water. Few students referred to refraction, while, during the case, they either attempted interpretations based on reflection or used a mental construction labeled 'vision rays', i.e., they assumed that light is transmitted from the eye to the object. In addition, several students simply were confined into making references to specific parts of the arrangement such as the water or the container. Bouwens' research also led to similar results [21].

In research conducted with high school and university students on a range of tasks related to refraction such as the propagation of light through prisms, concave, convex and flat surfaces or the formation of patterns, it was found that, very often, students were not able to successfully apply the law of refraction as a general scheme that enables them to deal with a number of different tasks in a stable way [22–25]. Thus, they usually dealt with experimental tasks focusing their attention on the external surface features of objects. Along the same research orientation, Kagawong et al. [26] studying 11- and 12-year-old students' representations of refraction asked them to create diagrams based on a simple experimental situation in which the path of light rays was traced. The results showed that the students could not apply the basic laws of refraction, even though they had been taught Geometric Optics.

While few studies have been conducted on the transformation of students' representations through systematic instruction, their results seem quite satisfactory. In an attempt to study the transformation of representations on refraction of 11-year-old students, data from two groups of students were compared. In particular, the experimental group worked with 'prediction–observation–explanation' (POE) inquiry-based learning model, while the control group worked with the traditional method. The data from this study showed that students in the experimental group responded well to qualitative interpretations of refraction-related phenomena such as observations with convex and concave lenses [27]. Moreover, the results in studies on 15-year-old students where the analogy of the light

that propagates from air to glass with a pair of wheels that change direction as they rolled diagonally from a hard to a soft surface was employed were satisfactory [28,29]. From the perspective of transforming student and teachers' representations of refraction, study documents and tasks based on students' misconceptions were used, aiming to lead to change [30]. Specifically, an experimental group worked with these texts, while a control group attended the traditional lessons. The data showed that students in the experimental group had better results in interpreting the phenomena of refraction. It should be noted that satisfactory results for the transformation of student–teachers' representations were also recorded when they both engaged in inquiry-based teaching processes [31]. In a different direction, a study compared the results of two teaching interventions for concepts and phenomena in optics, including refraction. The first teaching intervention was conducted through a phenomenological theoretical framework, while the second was through a classical model-based approach. The data of this study showed that both the students' performance on refraction, as well as their interest, were better when using the phenomenological framework than the model-based approach [32,33].

It is a fact that the research conducted in recent decades on teaching and learning refraction has been quite limited. It is usually carried out with older students, while the tasks that entail concave and convex lenses, as well as prisms, have proven to be complex, since they create difficulties in exploiting the refraction of light. In addition, difficulties arise in applying Snell's law, which, in fact, has a rather empirical rather than interpretative character. In order to gain a deeper understanding of these difficulties, we tried to study the representations and schemata of grade 9 students who have already been taught refraction. That is, we studied children's representations and schemata of Snell's law, which refers to the relationship between the angles of reflection and refraction and is considered to be appropriate for this education level.

## 2. Materials and Methods

### 2.1. Research Questions and Tools of Analysis

Our research questions aim to capture, in a systematic way, the basic elements that organize and constitute the representations and schemata of 9th grade students about the phenomenon of refraction. The term mental schema refers to a group of representations that express an organized pattern of students' thoughts about the phenomenon under study. Therefore, in the current study, we aspire to explore the following two fundamental issues regarding children's thinking about refraction:

1. Identify students' representations.
2. Reveal how students' representations are grouped and how they relate to the schemata.

To fulfill the aim of the research, we relied on an initial descriptive analysis [34] based on the children's responses to the individual tasks concerning light refraction. The children's responses for each task were qualitatively categorized, and thus, individual student's mental representations of refraction were detected. A multivariate analysis was then performed, through which the individual representations were grouped into coherent sets that identified the students' individual schemata about refraction.

A multivariate analysis constitutes a suitable set of methods to investigate relationships between variables, especially when the research data concern simultaneous measurements of many parameters [35]. Multivariate analysis methods include different potentials, such as (1) data reduction to the degree that the situation under study can be represented as simply as possible, (2) sorting and grouping numerical variables or categories of qualitative variables in order to investigate similarities and dissimilarities between groups, (3) investigation of dependence and/or interdependence relations among variables or categories, and (4) predictions of relationships between variables or categories.

In our analysis, we used the Multiple Correspondence Analysis, which is suitable for analyzing categorical data. The Multiple Correspondence Analysis [36,37] was used for analyzing globally the students' responses to research tasks (open questions based on an activity sheet). This method was used because a descriptive statistical analysis

of the students' responses showed only their different approaches to the various tasks. To overcome the limitations above, we used the multivariate analysis to further analyze our research data and obtain an overall view of the students' representations and reveal numerous correlations across the research questions.

Our multivariate analysis aims at a deeper investigation of students' cognitive schemata of the refraction phenomenon after they were taught about it in school. We have chosen this kind of analysis as it can aid us in revealing the various correlations, as well as studying the students' reasoning thoroughly. In this case, grouping the representations into sub-groups provides a more comprehensive picture of how they are organized into students' explanatory cognitive schemata of light refraction.

### 2.2. Sample

The sample of the survey was 213 students (102 boys and 111 girls) aged 14–15 years old from 4 Greek schools located in areas resided in by parents of all socioeconomic levels (low, middle and high). The students were recruited from 10 different classes of the 9th grade (3rd grade of Greek secondary school). They had been taught some basic lessons about refraction in primary school and a few applications of Snell's law in the second year of secondary school. The sample of the study consisted of students who attended the third grade of secondary school. According to the Greek curriculum, physics is taught in this grade for two hours per week. The chapter on optics, which is the last chapter of the physics textbook, deals mainly with refraction and reflection, as well as convex and concave mirrors. These phenomena are taught in a semiquantitative manner through Snell's law and textbook exercises. Experimental exercises are confined on apparatuses such as a Newton disc, also known as the disappearing color disc, and a dispersive prism. It should be noted that the phenomenon of refraction had already been taught in the sixth grade of primary school in a qualitative manner. Written consent was asked and obtained from all students and their parents to participate in the survey.

### 2.3. Tasks

The children's mental representations and cognitive schemata were studied through an activity sheet parted by 3 diverse tasks containing open questions. In each task, the students were asked to predict the path of a light ray that strikes on the smooth, flat surface of two media and justify their view. The activity sheet was completed individually and anonymously. In what follows, these three tasks are presented.

#### 2.3.1. Task 1

A thin beam of light propagates through air and strikes sideways into the smooth and flat surface of a glass tile (see Figure 1). Please draw the path of the ray through the tile and justify your choice.

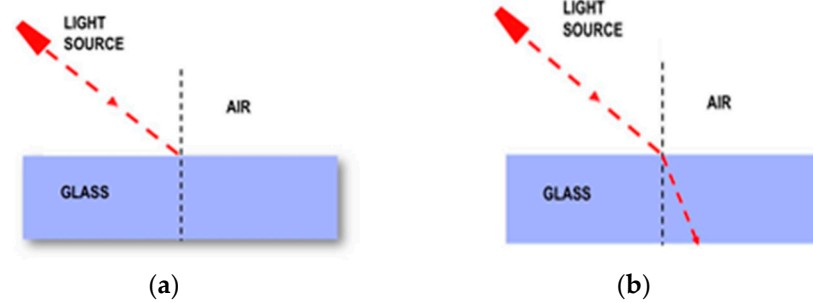

|     |     |
|:---:|:---:|
| (**a**) | (**b**) |

**Figure 1.** The movement of the light as it propagates through air and strikes into the glass tile ((**a**) depicts the figure that should be filled in by students, and (**b**) depicts the answer that is consistent with Geometric Optics).

### 2.3.2. Task 2

The light source is located within the water, as shown in Figure 2. Please draw the path of the ray and justify your choice.

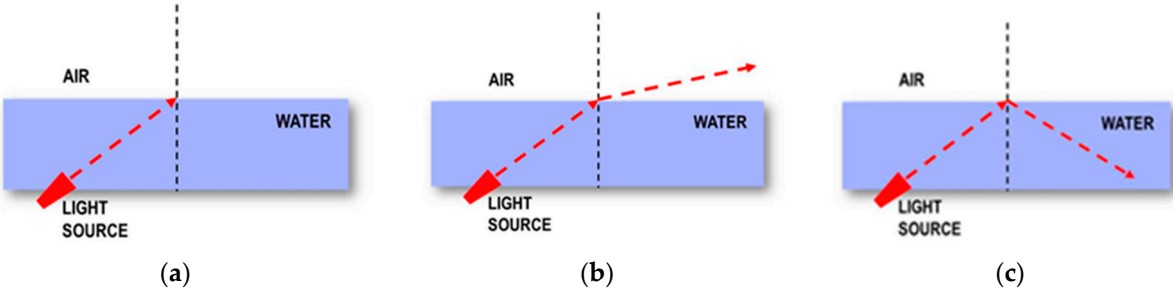

(**a**) (**b**) (**c**)

**Figure 2.** The movement of the light as it propagates from water to air ((**a**) depicts the figure that should be filled in by students, and (**b**,**c**) depict the answers that are consistent with Geometric Optics).

In this task, students were asked to draw the path of the light both within and out of the water and justify their predictions. The light moves from a dense transparent medium to a thinner one. It was clearly stated to students that the light source is located within the water and that the light strikes sideways on the water–air surface. In the case that students predict that the light will not exit the water, they draw the beam only within the water area. The aim of this task was to test whether the students hold a basic understanding of the total reflection phenomenon, at least at a qualitative level.

### 2.3.3. Task 3

The light source is located perpendicular to the edge of the glass tile (see Figure 3). Please draw the path of the ray after striking the surface of the glass tile and justify your choice.

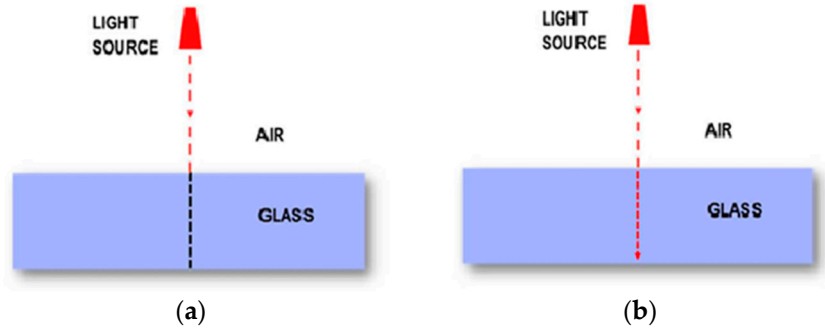

(**a**) (**b**)

**Figure 3.** The light strikes vertically from glass tile to air surface ((**a**) depicts the figure that should be filled in by students, and (**b**) depicts the answer that is consistent with Geometric Optics).

In this task, students were asked to draw the path of the light ray as it strikes vertically from glass tile to air surface.

## 3. Results

### 3.1. Students' Mental Representations Based on Different Tasks' Descriptions

An analysis of the student responses led to five diverse categories per task (i.e., 15 categories of analysis). The categorization of the representations was based on the combination of the prediction and justification given by the students in each task [34]. Sufficient justifications are defined as those that are consistent with the Geometric Optics model, i.e., the phenomenon of refraction explained based on Snell's law. In Task 2, there are two potentially correct answers related to the angle the light strikes the glass tile. In categorizing the answers, we considered as correct answers those that either predicted



refraction or total reflection, provided, however, that both of them were accompanied by a scientifically accepted justification. The frequencies of the responses for each representation category are presented in Table 1.

**Table 1.** Frequency of students' responses to the 3 tasks.

| Frequency of Students' Responses to Task 1 | Frequency | Percentage% |
|---|---|---|
| (a) The light ray will converge towards the vertical with sufficient justification (T1Correct) | 12 | 5.6 |
| (b) The light ray will converge towards the vertical with insufficient or no explanation (T1CorrectWithoutExplanation) | 15 | 7 |
| (c) The light ray will deviate from the vertical with insufficient or no justification (T1IncorrectDeviation) | 18 | 8.5 |
| (d) The light ray will continue to move in a straight line with insufficient or no justification (T1IncorrectStraightLine) | 123 | 57.7 |
| (e) The light ray will not enter the glass tile as it is reflected (T1IncorrectReflection) | 45 | 21.1 |
| **Frequency of students' responses to Task 2** | **Frequency** | **Percentage%** |
| (a) The light ray will deviate from the vertical with sufficient justification (T2Correct) | 13 | 6.1 |
| (b) The light ray will deviate from the vertical with no justification (T2CorrectWithoutExplanation) | 29 | 13.6 |
| (c) The light ray will converge towards the vertical with insufficient or no justification (T2IncorrectConvergence) | 21 | 9.9 |
| (d) The light ray will continue to move in a straight line with insufficient or no justification (T2IncorrectStraightLine) | 121 | 56.8 |
| (e) The light will not come out of the water (T2IncorrectReflection) | 29 | 13.6 |
| **Frequency of students' responses to Task 3** | **Frequency** | **Percentage%** |
| (a) The light ray will continue to move in a straight line with sufficient justification (T3Correct) | 20 | 9.4 |
| (b) The light ray will continue to move in a straight line with insufficient or no justification (T3CorrectWithoutExlpanation) | 89 | 41.8 |
| (c) The light enters the glass tile and is deflected from the vertical with insufficient or no justification (T3IncorrectDeviation) | 34 | 16 |
| (d) The light is reflected perpendicular to the glass tile in the opposite direction (T3IncorrectReflection) | 51 | 23.9 |
| (e) The light ray is reflected and diffused in different directions (T3IncorrectReflectionDeviation) | 19 | 8.9 |

Drawing from the data presented above, it was found that, in all three tasks, less than 1/10 students managed to construct representations compatible with Snell's law. Moreover, it was found that, while a small number of predictions seemed correct, they were actually intuitive, as they were insufficiently justified. In the vast majority of students' responses, they recorded unstable and contradictory answers regarding the issues of prediction and/or justification. This finding clearly highlights a strong difficulty in the construction of representations compatible with the Geometric Optics model. This fact certainly requires deeper investigation, as the difficulty in the phenomenon of refraction, to a great extent, is due to the nature of the model itself, which, instead of a qualitative interpretation, essentially refers to the technical mathematical application indicated by Snell's law.

### 3.2. Identification of Students' Cognitive Schemata about Refraction

The students' individual representations, as presented above, can be studied on a second level, forming more complex explanatory schemata through which students interpret the phenomenon of light refraction and its individual aspects. These schemata generally hold a specific logical organization in students' thinking. The study of these entities is of particular interest, since it provides, on the one hand, specific criteria for comparing them with scientific–school knowledge and, on the other hand, it offers a framework for a deeper analysis in organizing appropriate teaching interventions to transform them in case they are not compatible with the scientific–school knowledge.

To study the overall structure of students' schemata about light refraction, we applied the Multiple Correspondence Analysis (MCA) method. This analysis is a well-established multivariate method allowing us to analyze and describe graphically and synthetically a large amount of research data [36]. The MCA offers effective tools that can help us to overcome the intrinsic limitations of the descriptive statistics. It aims at the graphical representation of the structure of non-numerical (categorical) multivariate data. The leading principle of the MCA statistical method is that complex multivariate data can be accessible by displaying their main regularities and patterns in graphs and diagrams. The subjects under study are usually described by many categories. In our study, the three tasks (variables of analysis) are described by fifteen categories (Table 1). Therefore, the global structure of students' representations and schemata cannot be revealed through conventional statistical methods. With the help of the MCA, as an exploratory statistical method, we can derive not only students' representations but also how their schemata are structured across the various tasks. Furthermore, we can construct a topographic graph of those categories, thus making students' classification easily explainable based on their cognitive approach to the different tasks [38]. Finally, the detailed correlation between the fifteen categories of the three tasks that characterize the subjects of our study has a qualitative form, allowing us to construct valid assumptions offered for further study and analysis.

We performed our MCA using the statistical software SPAD version 7.4. According to Roux and Rouanet [39], SPAD is the most appropriate statistical software to conduct this kind of analysis. SPAD gives highly readable graphical representations of clouds of individuals and of categories. We applied the CORMU (Multiple Correspondence Analysis—MCA) method. As categories of the nominal variables, we used the various responses students gave to the three tasks (see Table 1). The MCA procured us a number of factors, which determined all the information produced. Each factor was described by two parameters [36]: (a) the eigenvalue $\lambda$, which corresponds to the eigenvectors describing the values of the variables implicated in the analysis, and (b) the coefficient of inertia $\tau$, which is the proportion of the total information in the factor, as it is provided by the MCA.

The MCA factors are categorized by descending order according to their importance, as far as the total information provided. Table A1 (Appendix A) presents the eigenvalues and the coefficients of inertia for the first six factors revealed by our analysis. They cumulatively represent the inertia at a percentage of 59.86%, which corresponds to 59.86% of the total information produced by our analysis. In our study, we analyze extensively the five first factors, which offer 51.56% (Table A1, Appendix A) of the total information but allow us to interpret 14 out of the 15 categories concerning students' representations. In is only the T2Incorrect category that does not appear in this analysis, as it appears for the first time in factor 6. For this reason, it is not necessary to extend our analysis to the lower order factors, since they can give only complementary information about the same categories.

The analysis presented above provides us with a number of interesting findings. Quite interestingly, the first three factors explain 12 out of the 15 categories of representations that emerged from our analysis, while the 3 remaining categories appear in the fourth, fifth and sixth factors. Most importantly, the data lead us to five main schemata concerning the refraction phenomenon. In what follows, we present the main characteristics of these schemata based on the weight of their occurrence in the context of the analysis carried out. At the second level, we explore in detail how these schemata are constituted.

### 3.2.1. Mental Schema MS1: 'Mental Schema of the Undisturbed Linear Path'

The mental schema MS1 is defined as the 'mental schema of the undisturbed linear path'. This schema is based on an assumption that does not take into account the refraction phenomenon at all. Particularly, it involves reasoning that does not recognize that the propagation of the light through the interface of two media of different densities implies the deflection of its path. Therefore, this scheme is not in the context of reasoning within Geometric Optics. A schematic representation of this schema across all three tasks is presented in Figure 4.

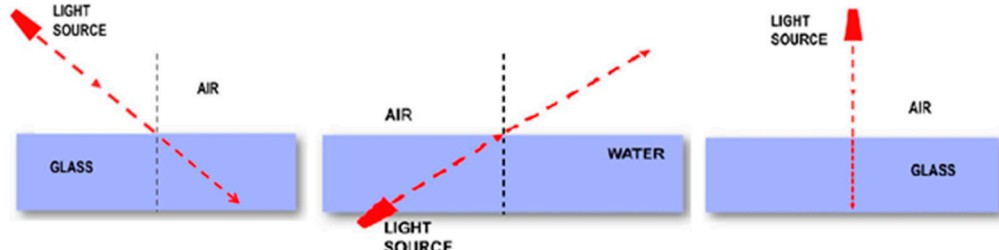

**Figure 4.** Typical representations belonging to the mental schema of the undisturbed linear path.

3.2.2. Mental Schema MS2: 'Mental Schema Compatible with the Geometric Optics Model'

The mental schema MS2 is described as a 'mental schema compatible with the Geometric Optics model'. It is a schema that generally leads to reasoning compatible with Snell's law. This mental schema appeared in the survey data with two variants (MS2a and MS2b), which are presented below. Typical para-examples of the representations belonging to this schema are presented in Figure 5.

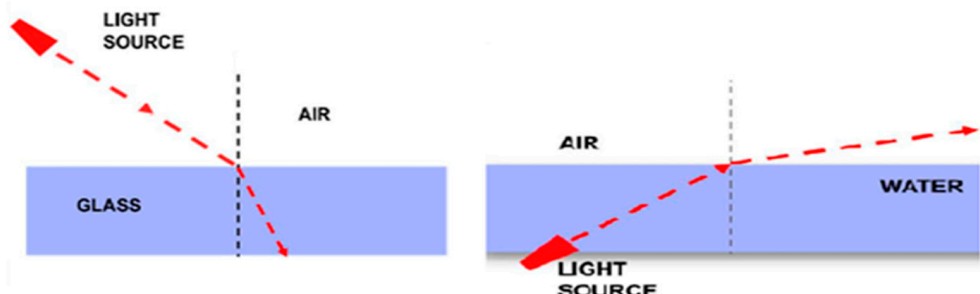

**Figure 5.** Typical representations belonging to the mental schema that are compatible with the Geometric Optics model.

3.2.3. Mental Schema MS3: 'Mental Schema Incompatible with the Geometric Optics Model'

The mental schema MS3 is described as 'a mental schema incompatible with the Geometric Optics model'. Students here formulate reasoning that is far from what the geometric perspective dictates. This mental schema appeared in the survey data with three variants (MS3a, MS3b and MS3c), which are presented below. A particular feature of this schema is that it was captured by responses confined in the first two tasks. Indicative representations belonging to this schema are shown in Figure 6.

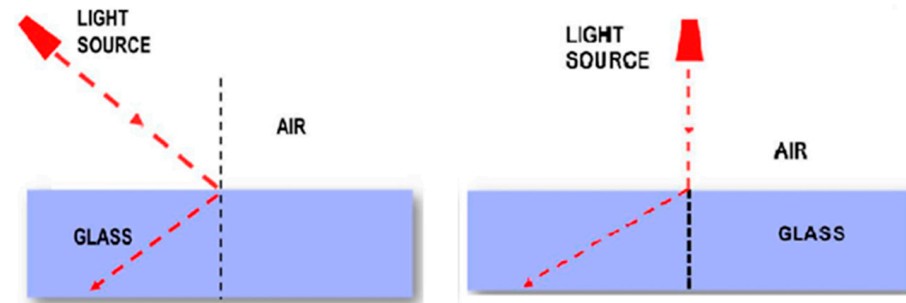

**Figure 6.** Typical representations belonging to the mental schema that are incompatible with the Geometric Optics model.

3.2.4. Mental Schema MS4: 'Mental Schema of the Deficient Approach to the Geometric Optics Model'

The mental schema MS4 is described as 'mental schema of the deficient approach to the Geometric Optics model'. Particular features of this schema are apparent in responses

that either tend to converge the light ray towards the vertical and/or are not clearly justified across all three tasks. Both of these features highlight an intuitive approach to the phenomenon of refraction that, despite its deficiencies, signals some initial approach to the Geometric Optics model. This mental schema appeared in the survey data with three variants (MS4a, MS4b and MS4c), which are presented below. Indicative representations belonging to this schema are shown in Figure 7.

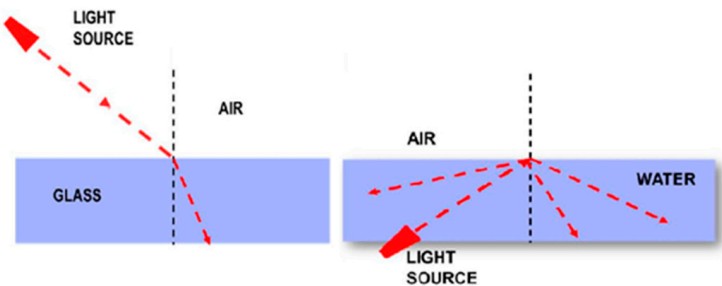

**Figure 7.** Typical representations belonging to the mental schema that deficiently approach the Geometric Optics model.

### 3.2.5. Mental Schema MS5: 'Mental Schema of Confusion between Refraction and Reflection/Diffusion'

The mental schema MS5 is described as the 'mental schema of confusion between refraction and reflection/diffusion'. This schema includes responses in which reflection seems to dominate the act of light ray as it propagates from one media to the other. It was mainly captured by responses confined in the first and third tasks and actually highlighted a group of students who seemed unable to conceptualize the phenomenon of refraction. Indicative representations belonging to this scheme are shown in Figure 8.

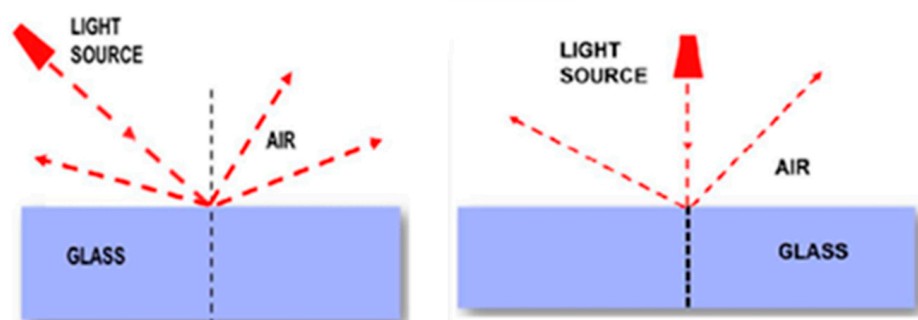

**Figure 8.** Typical representations belonging to the mental schema of confusion between refraction and reflection/diffusion.

In what follows, drawing from the statistical analysis, we present in detail the composition of these schemes. In particular, on the one hand, the results of the MCA are presented by describing and analyzing the individual factors. On the other hand, two factorial planes are presented that are constituted (a) by the first two most informationally important factors and (b) by the third and fifth factors. These factorial planes were chosen as they contained the set of schemata resulting from the analysis.

### 3.3. Structure of Students' Cognitive Schemata about Refraction

In this section, we present the structure of schemata about refraction as provided by the MCA based on the most important factors. Each factor (as an axis) gathers on each side one group of categories (students' responses). Therefore, each factor organizes and represents two schemata, which contain specific representations while highlighting the contradictions between these schemata.

### 3.3.1. First Factor: MS2 Versus MS3 Cognitive Schemata

The first factor has an eigenvalue $\lambda_1 = 0.4989$ and coefficient of inertia $\tau_1 = 12.47\%$ (Table A1, Appendix A). This is characterized as the well-structured mental schema (MS2a: the 'mental schema compatible with the Geometric Optics model') versus some fundamental alternative representations, which involve the reflection/deflection of light (MS3a: the 'mental schema incompatible with the Geometric Optics model'). It is a very important factor in our analysis, since it shows the contradictions between:

(1) the students' correct responses, which concern a well-defined mental schema, MS2a (one variant of the MS2 mental schema), with effectual reasoning in the first two tasks;

(2) some alternative representations exhibited by the majority of students in the sample, which were based on the idea that the light ray will either not enter the glass tile as it is reflected or diffused or will continue to propagate through glass tile moving in a straight line (MS3a).

It is evident that students' representations are explicitly grouped around two poles (Table A2, Appendix B). The first pole is defined by the students having a well-structured mental schema about the refraction of light, which is closer to the Geometric Optics model, despite any deficiencies.

This mental schema (MS2a: 'mental schema compatible with the Geometric Optics model', one variant of the MS2 mental schema) is determined by students who broadly show across all three tasks the following answers:

Task 1 (a): The light will converge towards the vertical with sufficient justification.
Task 2 (a): The light will deviate from the vertical with sufficient justification.
Task 3 (c): The light enters the glass tile and is deflected from the vertical with little or no justification.

The second pole (MS3a) is determined by those students who believe that:

Task 1 (e): The light will not enter the glass tile as it is either reflected or diffused.
Task 1 (d): The light will continue to move in a straight line with little or no justification.
Task 3 (d): The light is reflected perpendicular to the glass tile in the opposite direction.

In this case, we find a mental schema (MS3a, one variant of the MS3 mental schema) that is incompatible with the Geometric Optics model. We will find the MS3 schema again during the second factor analysis.

### 3.3.2. Second Factor: MS1 Versus MS3 Cognitive Schemata

The second factor has an eigenvalue $\lambda_2 = 0.4636$ and coefficient of inertia $\tau_2 = 11.59\%$ (Table A1, Appendix A). This factor is characterized by the contradiction between two alternative schemata (Table A2, Appendix B): MS1a ('mental schema of the undisturbed linear path', one variant of the MS1 mental schema) and MS3b ('mental schema incompatible with the Geometric Optics model', one variant of the MS3 mental schema). This is also a very important factor in our analysis, since it shows the contradiction between students who state across all three tasks that the light ray will move in a straight line and those who point out that the light ray will get deflected. Thus, the students' representations are grouped around two poles.

The first pole is determined by those students who have a well-structured alternative mental schema about the refraction of light. More specifically, the mental schema MS1a has students who broadly show across all three tasks the following answers:

Task 1 (d): The light will continue to move in a straight line with insufficient or no justification.
Task 2 (d): The light will continue to move in a straight line with insufficient or no justification.
Task 3 (b): The light will continue to move in a straight line with insufficient or no justification.

Drawing from the frequencies of the responses (Table 1), it appears that the strongest mental schema is that of the undisturbed straight line. Indeed, the majority of students seem to rely on this initial explanation schema.

The second pole (MS3b) is determined by those students who believe that:

Task 1 (e): The light will not enter the glass tile as it is reflected or diffused.
Task 3 (c): The light enters the glass tile and is deflected from the vertical with insufficient or no justification
Task 3 (d): The light is reflected perpendicular to the glass tile in the opposite direction.

As stated above, these are students that rely on MS3b: the light ray will be deflected either by reflection or refraction or by another kind of deflection that is not clearly explained (see Figure 6).

### 3.3.3. Third Factor: MS2 Versus MS4 Cognitive Schemata

The third factor has an eigenvalue $\lambda 3 = 0.3895$ and coefficient of inertia $\tau 3 = 9.74\%$ (Table A1, Appendix A). This factor is characterized by the contradiction between two alternative schemata (Table A2, Appendix B): MS2b ('mental schema compatible with the Geometric Optics model', one variant of the MS2 mental schema), which have already been found in the first factor, and MS4a ('mental schema of the deficient approach to the Geometric Optics model', one variant of the MS4 mental schema). It is a very important factor in our analysis, since it shows the contradiction between:

(1) students' correct responses (well-defined mental schema) with sufficient justification (MS2b);

(2) an alternative mental schema (MS4a), which was characterized as the 'mental schema of the deficient approach to the Geometric Optics model'. This schema was based on the idea that the light ray will either converge towards the vertical or continue to move perpendicular to the dividing surface without justifying this explanation.

Here, students' representations are also explicitly grouped around two poles. The first pole is defined by the students with a well-structured mental schema about the refraction of light that is almost compatible with the scientific model. This mental schema (MS2b) is determined by students who broadly show across all three tasks the following answers:

Task 2 (a): The light ray will deviate from the vertical with sufficient justification.
Task 2 (b): The light ray will deviate from the vertical with no justification.
Task 3 (a): The light ray will continue to move in a straight line with sufficient justification.

Although this mental schema, which is composed of answers given by the students in tasks 2 and 3 and corresponds to specific representations, is not identical in all its individual aspects with the corresponding mental schema that appears in the first factor (MS2a), it is almost compatible with the scientific model of light refraction.

The second pole (MS4a) of this factor is determined by those students who believe that the light rays converge to the vertical as soon as they do not move perpendicular to the contact surface of the two media. More specifically, the answers concern:

Task 1 (b): The light ray will converge towards the vertical with insufficient or no justification.
Task 2 (c): The light ray will converge towards the vertical with insufficient or no justification.
Task 3 (b): The light ray will continue to move in a straight line with insufficient or no justification.

These are students who rely on MS4a: the 'mental schema of the deficient approach to the Geometric Optics model'. The usage of this schema is often based on the convergence of the light ray to the vertical, either with insufficient or without justification at all.

### 3.3.4. Fourth Factor: MS3 Versus MS4 Cognitive Schemata

The fourth factor has an eigenvalue $\lambda 3 = 0.3652$ and coefficient of inertia $\tau 3 = 9.13\%$ (Table A1, Appendix A). This is characterized (see the two groups in Table A2, Appendix B) by one variant of mental schema MS3 (MS3c) and one variant of mental schema MS4 (MS4c).

The first pole of this factor (MS3c) is determined by those students who believe that:

Task 1 (e): The light will not enter the glass tile as it is reflected.
Task 3 (d): The light is reflected perpendicular to the glass tile in the opposite direction (T3IncorrectReflection).

The second pole of this factor (MS4c) is determined by those students who believe that light rays will change course without being able to reason their view according to the Geometric Optics model. More specifically, the answers concern:

Task 1 (b): The light ray will converge towards the vertical with insufficient or no justification.

Task 2 (b): The light ray will deviate from the vertical with no justification.

Task 3 (c): The light ray enters the glass tile and is deflected from the vertical with insufficient or no justification.

Both of these schemata have already been found in the previous axes; therefore, there is no need to analyze this axis further.

### 3.3.5. Fifth Factor: MS4 Versus MS5 Cognitive Schemata

The fifth factor has an eigenvalue $\lambda 5 = 0.3454$ and coefficient of inertia $\tau 5 = 8.63\%$ (Table A1, Appendix A). This factor is characterized by the contradiction between two alternative schemata (Table A2, Appendix B): MS4 ('mental schema of the deficient approach to the Geometric Optics model'), which were already found in the third factor, and MS5 ('mental schema of confusion between refraction and reflection/diffusion'). This is an important factor in our analysis, since it shows the contradiction between:

(1) an alternative mental schema, which we called the mental schema of the deficient approach to the Geometric Optics model (MS4). Here, we find a variant (MS4b), which was based on the idea that the light ray will either converge to the perpendicular or continue to move vertically without justifying these justifications;

(2) another alternative scheme, which we called the mental schema of confusion between refraction and reflection/diffusion (MS5: 'mental schema of confusion between refraction and reflection/diffusion').

Here, students' representations are also explicitly grouped around two poles.

The first pole is defined by the students with the mental schema of the deficient approach to the Geometric Optics model (MS4b) and involves students who believe that light rays are either deflected or continue without deflection without being able to formulate clear justifications. Students generally give the following answers:

Task 2 (b): The light ray will deviate from the vertical with no justification.

Task 3 (b): The light ray will continue to move in a straight line with insufficient or no justification.

Task 3 (c): The light enters the glass tile and is deflected from the vertical with insufficient or no justification.

This mental schema, which is composed of answers given by students in tasks 2 and 3, primarily concerns 'the mental schema of the deficient approach to the Geometric Optics model' (MS4b). While the answers here are given mainly with insufficient or no justification at all, they tend to be close to the way of reasoning that governs the geometric perspective.

The second pole of this factor is determined by those students who confuse reflection with refraction (MS5: 'mental schema of confusion between refraction and reflection/diffusion'. More specifically, the answers concern:

Task 1 (c): The light ray will deviate from the vertical with insufficient or no justification.

Task 3 (e): The light ray is reflected and diffused in different directions.

### 3.4. Organization and Graphical Representation of Students' Cognitive Schemata about Refraction

In this section, we present two factorial planes: (a) the one constituted by the first two—and most important in terms of information—factors and (b) the one constituted by the third and fifth factors. These are the factorial planes comprising the set of schemata that came up in our data analysis. In particular, in these factorial planes are placed the subgroup categories of the representations. In this way, we can identify and explain graphically how students' representations are organized and structured and point out how, at the same time, they are constituted in the schemata while highlighting the contradiction between these schemata.

### 3.4.1. First Factorial Plane (Factors 1 and 2): MS1, MS2 and MS3 Cognitive Schemata

An examination of the first factorial plane (Figure 9), which is formed by the first two factors, provides interesting insights into the way in which students' basic schemata of light refraction are organized. The first two factors were chosen, because they represent the most important information of the MCA. On the factorial plane, we observe three groups of categories, i.e., answers to the three tasks, which express subsequent students' mental representations. As already presented, each group encompasses a main mental schema for light refraction.

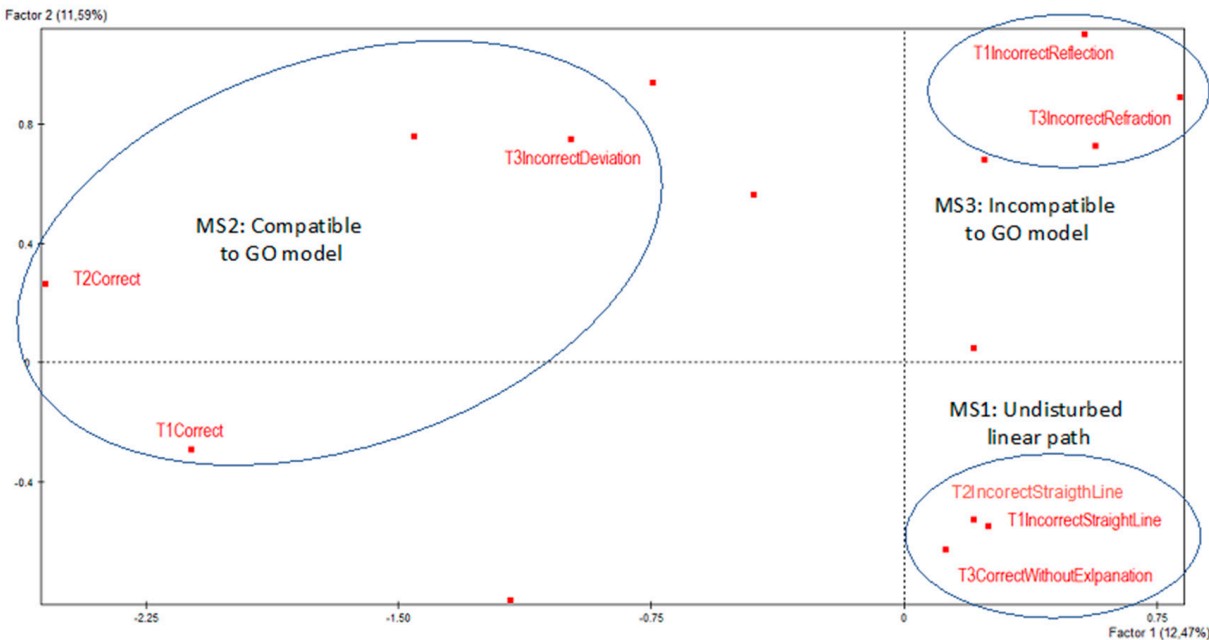

**Figure 9.** Graphical representation of students' cognitive schemata about light refraction (schemata MS1, MS2 and MS3). The factorial plane is defined by the first two factors.

The first group counts as the 'mental schema of the undisturbed linear path' (MS1—variant MS1a) and is represented in the lower right quadrant of the factorial plane in a three-point cloud. The second group counts as the 'mental schema compatible with the Geometric Optics model' (MS2—variant MS2a) and is represented by the three-point cloud in the left-hand quadrant of the factorial plane. The third group counts as the 'cognitive schema incompatible with the Geometric Optics model' (MS3—variant MS3c) and is represented in the upper-right quadrant. The factorial plan thus represents the three main schemata resulting from our data analysis. Any additional information that the factorial plane provides us is the following: Due to the way the MCA operates, the closer to the beginning of the axes of the factorial plane a cloud is placed, the more subjects in the survey exhibit the mental schema represented by this cloud. Consequently, as MS2 is the most compatible with the Geometric Optics model, it is a mental schema that fewer subjects in the sample possess compared to the MS1 and MS3 schemas. This finding is already evident in the frequency tables (Table 1) with respect to the individual representations that form these mental schemas.

### 3.4.2. Second Factorial Plane (Factors 3 and 5): MS2, MS4 and MS5 Cognitive Schemata

The examination of the factorial plane (Figure 10), which is formed by the third and fifth factors, provides additional information about the way in which the students' last two schemata of light refraction are organized. These are schema MS4 ('mental schema of the deficient approach to the Geometric Optics model') and schema MS5 ('mental schema of confusion between refraction and reflection/diffusion'). We chose this factorial plane of the MCA, because it represents information that is not provided by the previous

factors regarding mental schema MS5 ('mental schema of confusion between refraction and reflection/diffusion') and is the only category that has not been described so far by our analysis. This factorial plane also represents some categories concerning mental schema MS4 (variant MS4a) ('mental schema of the deficient approach to the Geometric Optics model') that do not appear in the previous factorial plane (Figure 9), as well as some categories concerning schema MS2 (variant MS2b) ('mental schema compatible with the Geometric Optics model').

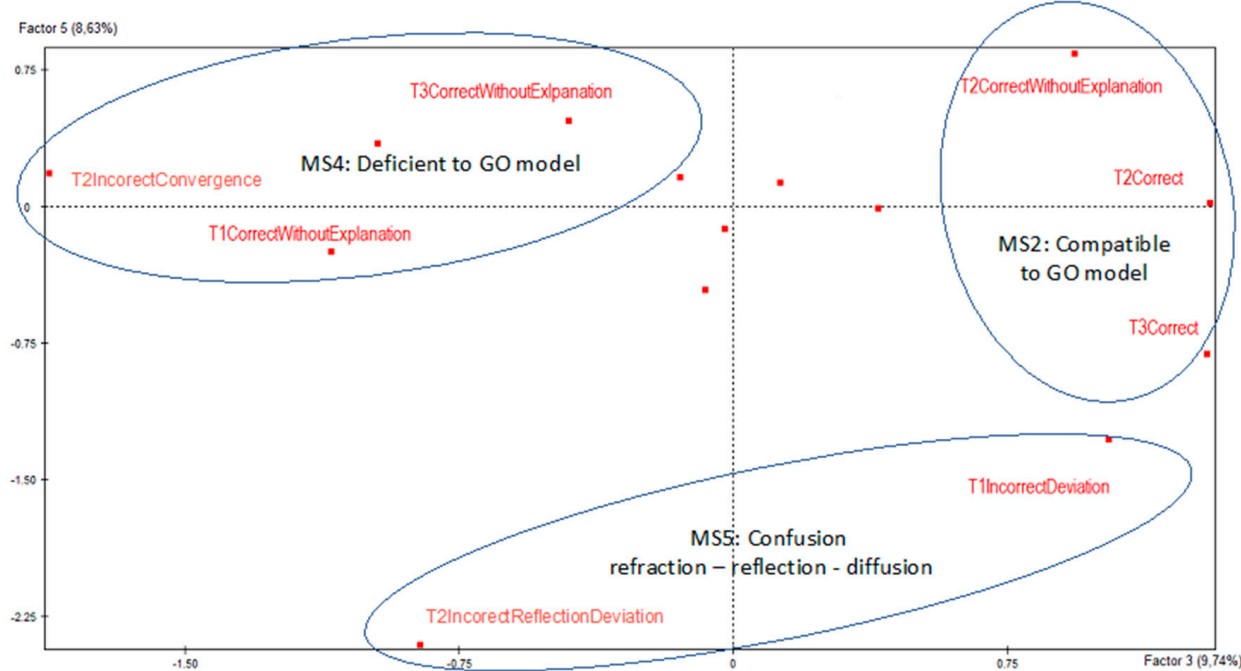

**Figure 10.** Graphical representation of students' cognitive schemata about light refraction (schemata MS2, MS4 and MS5). The factorial plane is defined by the third and fifth factors.

In particular, in the factorial plane, we observe three main groups of categories. The first group contains the two categories mentioned earlier and concerns the 'mental schema regarding the confusion between refraction and reflection/diffusion' (MS5). The second group concerns the 'mental schema of the deficient approach to the Geometric Optics model' (MS4) (variant MS4a), while the third group contains categories concerning mental schema MS2 (variant MS2b). It can be seen that one variant of schema MS2 appears in these factors as well. This is due to the nature of the MCA; each factor counts for specific categories of the analysis and represents a percentage of the total information, while these categories can be found on more than one axis.

### 4. Discussion and Conclusions

In the current research study, we attempted to explore, on the one hand, the mental representations of 9th grade students about refraction and, on the other hand, to study how these representations are grouped together and construct more stable forms of thinking called schemata.

The results of the descriptive analysis captured the individual representations of the students as they appeared in the three tasks. The findings for these representations, which were classified into 15 distinct categories, are in line with the existing relevant literature. Difficulty in approaching the refraction phenomenon was found in all tasks, while more than half of the students were not able to deal with the refraction phenomenon at all [22–25]. Given the fact that the students in the sample had already been taught the refraction phenomenon, these data clearly highlight the issue of the ineffectiveness of traditional instruction for teaching light propagation. However, we have to take into

account that the approach to this teaching was model-based [40–42]. In recent years, the emergence of a phenomenological perspective might perhaps give new dimensions to the teaching and learning of refraction [32,33].

The results of the Multiple Correspondence Analysis (MCA) allowed the approach of the organization and constitution of the 15 categories of mental representations in order to establish the constitution of the schemata, i.e., explanatory mental entities for the concept of refraction. These schemata hold interesting elements that clearly allow for further deeper thinking.

The strongest mental schema (MS1) in terms of the frequency of occurrence in the survey sample is the one that recognizes that light rays move in a straight line without deflection, regardless of the change of the medium of propagation. Clearly, this is a thought pattern that has not been affected at all by school instructions. In addition, refraction seems to be out of context for those students who created their answers based on MS5, as they predicted reflection for tasks in which the phenomenon of refraction was apparent. This mental schema counted as a basic misconception that emerged in the reasoning of a small number of students and is apparently distant from school–scientific knowledge.

Regarding the other three schemes (MS2, MS3 and MS4), the refraction effect seems to be recognized in various ways. In particular, the reasonings found in the MS3 scheme are not governed by fixed rules. Here, the students sometimes refer to reflection and sometimes to an undisturbed path while other times pointing out a kind of change in the path of the light ray. In all these reasonings, there is inadequate or a total absence of justification. Consequently, in this scheme, the changes in the predictions are continuous and without any regularity, so, in fact, it is a pseudo-schema that consists of representations that are not related to the Geometric Optics model.

The representations that comprise the MS4 schema are characterized by a basic pattern of scientifically correct predictions with insufficient justifications. It seems that it is a schema that, while acting as the initial phase of dealing with the Geometric Optics model, is still unstable and intermittent and therefore leads to difficulties in conceptualizing the phenomenon of refraction.

In contrast, the usage of the MS2 schema compatible with Geometric Optics generally leads to correct predictions and justifications, although, in some cases, there are some ambiguities in the justifications. This schema does not seem to be very strong in terms of the frequency of occurrence, and certainly, our findings raise the issue of teaching strategies for the concept of refraction at this level of education.

The above-mentioned findings raise the matter of efficient teaching practices, as it is apparent that the different ways in which children deal with the concept of refraction through schemata have patterns that are relatively strong and stable. This is, after all, the precise meaning of mental schema. Nevertheless, the fact that the reasoning derived from these schemata is stable emphasizes the need for systematic and specifically oriented teaching interventions that go beyond both the traditional teaching of refraction, i.e., the presentation of the phenomenon and the declarative learning of Snell's law, in terms of both content and orientation. In addition, the inefficiency of a traditional didactic approach to refraction is pervasive in the results of our research. Therefore, moving towards this perspective, we are trying to design new learning environments that take into account the findings of the current study.

**Author Contributions:** Conceptualization, G.F. and K.R.; methodology, all authors; validation, K.R., V.K. and G.K.; formal analysis, all authors; investigation, G.F. and K.R.; resources, G.F., K.R. and V.K.; writing—original draft preparation, all authors; writing—review and editing, all authors. All authors have read and agreed to the published version of the manuscript.

**Funding:** This research received no external funding.

**Institutional Review Board Statement:** The study was conducted according to the guidelines of the Declaration of Helsinki, and approved by the Ethics Committee of Department of Educational Sciences and Early Childhood Education, University of Patras (protocol code No: 5/23.11.2021).

**Informed Consent Statement:** Informed consent was obtained from all subjects involved in the study.

**Data Availability Statement:** The study was conducted according to the guidelines of the Declaration of Helsinki and approved by Removed for peer review.

**Conflicts of Interest:** The authors declare no conflict of interest.

## Appendix A

**Table A1.** MCA parameter values.

| Factor | Eigenvalue λ | Coefficient of Inertia τ | Cumulated Percentage |
|:---:|:---:|:---:|:---:|
| 1 | 0.4989 | 12.47 | 12.47 |
| 2 | 0.4636 | 11.59 | 24.06 |
| 3 | 0.3895 | 9.74 | 33.80 |
| 4 | 0.3652 | 9.13 | 42.93 |
| 5 | 0.3454 | 8.63 | 51.56 |
| 6 | 0.3320 | 8.30 | 59.86 |

## Appendix B

**Table A2.** Five first factors' descriptions by active categories of MCA and cognitive schemata's description.

| Printout on Factor 1 by the Active Categories | | | |
|:---|:---|:---:|:---:|
| **Variable Label** | **Category Label** | **Test Value** | **Weight** |
| MS2a: "mental schema compatible with the Geometric Optics model" | | | |
| Task 1 | T1Correct | −7.52 | 12,000 |
| Task 2 | T2Correct | −9.46 | 13,000 |
| Task 3 | T3IncorrectDeviation | −6.26 | 34,000 |
| MS3a: "mental schema incompatible with the Geometric Optics model" | | | |
| Task 1 | T1IncorrectReflection | 4.06 | 45,000 |
| Task 1 | T1IncorrectStraightLine | 4.28 | 123,000 |
| Task 3 | T3IncorrectReflection | 6.69 | 51,000 |
| **Printout on factor 2 by the active categories** | | | |
| **Variable label** | **Category label** | **Test Value** | **Weight** |
| MS1a: "mental schema of the undisturbed linear path" | | | |
| Task 1 | T1IncorrectStraightLine | −9.35 | 123,000 |
| Task 2 | T2IncorrectStraightLine | −8.81 | 121,000 |
| Task 3 | T3CorrectWithoutExlpanation | −7.73 | 89,000 |
| MS3b: "mental schema incompatible with the Geometric Optics model" | | | |
| Task 1 | T1IncorrectReflection | 8.31 | 45,000 |
| Task 3 | T3IncorrectDeviation | 4.76 | 34,000 |
| Task 3 | T3IncorectReflectionDeviation | 7.27 | 51,000 |

**Table A2.** *Cont.*

| **Printout on factor 3 by the active categories** | | | |
|---|---|---|---|
| **Variable label** | **Category label** | **Test Value** | **Weight** |
| MS2b: "mental schema compatible with the Geometric Optics model" | | | |
| Task 2 | T2Correct | 4.85 | 13,000 |
| Task 2 | T2CorrectWithoutExplanation | 5.40 | 29,000 |
| Task 3 | T3Correct | 6.07 | 20,000 |
| MS4a: "mental schema of the deficient approach to the Geometric Optics model" | | | |
| Task 1 | T1CorrectWithoutExplanation | −4.41 | 15,000 |
| Task 2 | T2IncorrectConvergence | −9.01 | 21,000 |
| Task 3 | T3CorrectWithoutExlpanation | −5.52 | 89,000 |
| **Printout on factor 4 by the active categories** | | | |
| **Variable label** | **Category label** | **Test Value** | **Weight** |
| MS4c: " mental schema of the deficient approach to the Geometric Optics model" | | | |
| Task 1 | T1CorrectWithoutExplanation | −6.64 | 15,000 |
| Task 2 | T2CorrectWithoutExplanation | −7.19 | 29,000 |
| Task 3 | T3IncorrectDeviation | −4.73 | 34,000 |
| MS3c: "mental schema incompatible with the Geometric Optics model" | | | |
| Task 1 | T1IncorrectReflection | 4.77 | 45,000 |
| Task 3 | T3IncorrectReflection | 4.48 | 51,000 |
| **Printout on factor 5 by the active categories** | | | |
| **Variable label** | **Category label** | **Test Value** | **Weight** |
| MS5: "mental schema of confusion between refraction and reflection/diffusion" | | | |
| Task 1 | T1IncorrectDeviation | −5.66 | 18,000 |
| Task 3 | T3IncorrectReflectionDeviation | −11.00 | 19,000 |
| MS4b: "mental schema of the deficient approach to the Geometric Optics model" | | | |
| Task 2 | T2CorrectWithouExplanation | 4.86 | 29,000 |
| Task 3 | T3CorrectWithoutExlpanation | 5.81 | 89,000 |
| Task 3 | T3IncorrectDeviation | 3.90 | 34,000 |

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
