# Peer review of "Mental Representations and Cognitive Schemata of Ninth Grade Students for the Refraction of Light"

_education, doi:10.3390/educsci13050467_

Round 1
Reviewer 1 Report
The research that has been carried out is extremely interesting, however, I have some lack of information about the research method. The research took into account justification statements and how students drew. These measures raise many questions relating to the evaluation of the drawings.
Section 2.3 Assignments 1, 2 and 3 were drawing tasks in which the student had to mark the line of a light ray. It was the direction of this ray that was being assessed. As I understand it, the angle of deviation from the dashed line was taken into account? Were rulers and protractors available to the students? Was a range of mistake taken into account when drawing? What was the range of error.
Item 2.3.2 As there are two correct options in task 2 how was the situation assessed when a student drew one of the correct answers? Did these happen?
Gender was not included among the variables because there were very few girls in the study group (11 out of 102 boys). However, this fact is quite important for the construction of conclusions. This is because it means that the results predominantly refer to boys, who may respond differently to tasks of a geometric nature. This is particularly relevant when we talk about "specific logical organization in students' thinking" (line 223).
The last sentence of the article, which speaks of a "specially designed digital learning environment", is extremely interesting. It would be good to expand it to indicate a course of action or remove it as going far beyond the topic addressed in the article.
Author Response
We are very grateful to Reviewer 1 for his very constructive and timely comments.
We have tried to work systematically to respond to all of them.
|
Reviewer 1 |
Authors |
|
1. The research took into account justification statements and how students drew. These measures raise many questions relating to the evaluation of the drawings. Section 2.3 Assignments 1, 2 and 3 were drawing tasks in which the student had to mark the line of a light ray. It was the direction of this ray that was being assessed. As I understand it, the angle of deviation from the dashed line was taken into account? Were rulers and protractors available to the students? Was a range of mistake taken into account when drawing? What was the range of error. Item 2.3.2 As there are two correct options in task 2 how was the situation assessed when a student drew one of the correct answers? Did these happen?
|
Thank you for this comment! All students participated in the study were provided with a ruler and a protractor in order to mark the lines in the drawings.
Regarding Item 2.3.2, as it is stated in the article ‘The categorization of representations was based on the combination of prediction and justification given by students in each task’ (p. 5). Therefore, in Task 2 we considered as correct answers those that suggest at least one correct option which, however, was necessarily followed by a scientifically accepted justification.
To make our point clearer, in the revised manuscript we added the following text:
p. 6, lines 226-230 – In Task 2 there are two potentially correct answers related to the angle the light strikes into the glass tile. In categorizing the answers, we considered as correct answers that either predicted refraction or total reflection, provided however that both of them were accompanied by a scientifically accepted justification.
|
|
2. Gender was not included among the variables because there were very few girls in the study group (11 out of 102 boys). However, this fact is quite important for the construction of conclusions. This is because it means that the results predominantly refer to boys, who may respond differently to tasks of a geometric nature. This is particularly relevant when we talk about "specific logical organization in students' thinking" (line 223). |
We would like to apologize as due to a typographical error we mistakenly wrote in the text that11 girls participated in the study, whereas in fact there were 111 girls. It goes without saying that we fixed that mistake in the revised manuscript. The issue of differences between boys and girls does not appear in the current study as in all the statistical tests we carried out there was neither a statistically significant nor a noticeable difference among the boys and girls group.
|
|
3. The last sentence of the article, which speaks of a "specially designed digital learning environment", is extremely interesting. It would be good to expand it to indicate a course of action or remove it as going far beyond the topic addressed in the article. |
Thank you for this comment. As it is a fact that this sentence goes beyond the topic addressed in the study, we decided in the revised manuscript to delete this reference and reformulate the last sentence.
|

Reviewer 2 Report
I like the theme of this article. However, I have the following concerns which require major revisions:
1) The authors use the terms "mental representations" and "cognitive schemata". As far as I can see, however, no definitions are provided for these terms. As a reader I would require them to follow the article.
2) In physics education literature, the terms student conceptions, learning difficulties or mental models are more widespread. Hence, please check if you need to stay with the above terms. If so, justify your choice, please.
3) The methods section should be improved. The authors state to use Multiple Correspondence Analysis without further describing their procedure. In the results section, however, they only talk about "basic factor analysis" (what is that? exploratory?) and report numbers in a way uncommon for reporting factor analysis results (while the rest is in the appendix). Please present statistical results the way it is commonly done in the body text and check your dataset for appropriatness for factor analysis.
4) The quality of the figures is very low. I suggest the authors to export their figures as pdf and to import that into their manuscript in order to ensure high quality.
5) From my opinion, the authors themselves mix up model and reality when they talk about light rays in ways such as "predict that the beam will not exit the wate". I would suggest to re-phrase all these text parts and to rather talk about "that the light will not exit the water" (and similarly throughout the manuscript).
6) In general, the results are rather weak: There are a lot of numbers for that the result is that there are students who hold cognitive schemata (in-)compatible with the scientifitc view plus a further one. However, readers do not get more in-depth insights into qualitative results. The more important would be, to discuss your results against the background of recent findings, in particular regarding student learning about light refraction. I suggest the authors to consider the follpwing very recent articles that they have not taken into account so far:
a) Fliegauf, K., Sebald, J., Veith, J. M., Spiecker, H., & Bitzenbauer, P. (2022). Improving Early Optics Instruction Using a Phenomenological Approach: A Field Study. Optics, 3(4), 409–429.
b) Sebald, J., Fliegauf, K., Veith, J., Spiecker, H., & Bitzenbauer, P. (2022). The World through My Eyes: Fostering Students’ Understanding of Basic Optics Concepts Related to Vision and Image Formation. Physics, 4(4), 1117–1134.
I am looking forward to reading the revised version of this article.
Author Response
We are very grateful to Reviewer 2 for his very constructive and timely comments.
We have tried to work systematically to respond to all of them.
|
Reviewer 2 |
Authors |
|
1. The authors use the terms "mental representations" and "cognitive schemata". As far as I can see, however, no definitions are provided for these terms. As a reader I would require them to follow the article. |
Thank you for this comment. Indeed, it is a fact that the terms of ‘mental representations’ and ‘cognitive schemata’ appear in the literature with diverse approaches and theoretical perspectives. In the current study, we adopted the approaches that meet the needs of the construction of physics concepts in children's minds as they focus on both the learning process and the construction of knowledge.
In the revised manuscript we added the following text
p. 2, lines 55-67 - The term ‘mental representation’ was chosen among others such as ‘alternative idea’, ‘misconception’ etc. This schematization of naïve entities of children's thinking, signifies not only a specific knowledge content, but an entity included into a cognitive system with concrete structures, ‘processing and mapping’ procedures. In that kind of systems, the term of "mental representations" is considered more suitable since it approaches the structural parameters as well as the functional interrelations of a wider system (Hubbard, 2007). In the same perspective, the concept of ‘cognitive schema’ was chosen. Based on Piaget's (1950) approach, the cognitive schema of a mental operation is its general structure, that is, what is kept unchanged in repetitions of the same operation, adapted to different situations and stabilized with practice. The cognitive scheme consists of organized elements of different past experiences and situations that form a relatively cohesive and persistent frame of knowledge which guides one's subsequent perception and consideration of the world (Segal, 1988).
Reference list
Hubbard, T. L. (2007). What is Mental Representation? And how does it relate to consciousness? Journal of Consciousness Studies, 14(1-2), 37-61. Piaget, J. (1952). The origins of intelligence in children. International University Press: New York, USA. Segal, Z. (1988). Appraisal of the self-schema: construct in cognitive
|
|
2. In physics education literature, the terms student conceptions, learning difficulties or mental models are more widespread. Hence, please check if you need to stay with the above terms. If so, justify your choice, please. |
Unfortunately, it is a fact that these terms, along with several others (i.e. misconceptions, alternative ideas, perceptions etc.) are used in diverse ways and in fact without a rule that indicates which term should be used each time. Therefore, in the present study, we chose to use the terms ‘mental representations’ and ‘cognitive schemata’ which have a clear semantic withing diverse fields of Psychology that deal with learning and Science education in general.
|
|
3. The methods section should be improved. The authors state to use Multiple Correspondence Analysis without further describing their procedure. In the results section, however, they only talk about "basic factor analysis" (what is that? exploratory?) and report numbers in a way uncommon for reporting factor analysis results (while the rest is in the appendix). Please present statistical results the way it is commonly done in the body text and check your dataset for appropriateness for factor analysis. |
Multiple correspondence analysis (MCA) is a statistical technique based on J.-P. Benzecri work for nominal categorical data, used to detect and represent underlying structures in a data set. MCA can be viewed as an extension of simple correspondence analysis (CA) in that it is applicable to a large set of categorical variables. MCA is an inductive approach to data analysis. Le Roux & Rouanet (2010), designate this statistical approach, including the analysis of structured data and its inductive extensions, geometric data analysis (GDA). To perform our MCA in a user-friendly way, we used the SPAD (release 7.4) software. The user can obtain eigenvalues, modified rates, coordinates, contributions, supplementary elements and test-values for both regular and specific MCAs. All results are returned in the Excel environment. This means that it is very easy to make additional computations on the results. Furthermore, the graphical module is very easy to use. It gives highly readable graphical representations of clouds of individuals and of categories.
Within the text, to make the presentation of the analysis easier to understand, we have replaced the phrase “We conducted our analysis using the statistical software SPAD version 7.4.” with the paragraph “We performed our MCA using the statistical software SPAD version 7.4. According to Roux and Rouanet (2010) SPAD is the most appropriate statistical software to conduct this kind of analysis. SPAD gives highly readable graphical representations of clouds of individuals and of categories” (p. 7, lines 276-279). Also, the phrase “In this section we present the structure of schemata about refraction as provided by MCA based on the basic factor analysis” has been replaced by the phrase “In this section we present the structure of schemata about refraction as provided by MCA based on the most important factors” (p. 10, lines 374-375).
Reference
Le Roux, B., & Rouanet, H. (2010). Multiple Correspondence Analysis. SAGE Publications, Inc.
|
|
4. The quality of the figures is very low. I suggest the authors to export their figures as pdf and to import that into their manuscript in order to ensure high quality. |
Thank you for this comment which clearly improves the readability of the article. If the article is accepted, in the final version of it we are going to convert the images to pdf.
|
|
5. From my opinion, the authors themselves mix up model and reality when they talk about light rays in ways such as "predict that the beam will not exit the water". I would suggest to re-phrase all these text parts and to rather talk about "that the light will not exit the water" (and similarly throughout the manuscript). |
Thank you for this comment. In the revised manuscript we decided to keep up with ‘reality’ and therefore we rephrase accordingly all the relevant parts.
|
|
6. In general, the results are rather weak: There are a lot of numbers for that the result is that there are students who hold cognitive schemata (in-)compatible with the scientific view plus a further one. However, readers do not get more in-depth insights into qualitative results. The more important would be, to discuss your results against the background of recent findings, in particular regarding student learning about light refraction. I suggest the authors to consider the following very recent articles that they have not taken into account so far: a) Fliegauf, K., Sebald, J., Veith, J. M., Spiecker, H., & Bitzenbauer, P. (2022). Improving Early Optics Instruction Using a Phenomenological Approach: A Field Study. Optics, 3(4), 409–429. b) Sebald, J., Fliegauf, K., Veith, J., Spiecker, H., & Bitzenbauer, P. (2022). The World through My Eyes: Fostering Students’ Understanding of Basic Optics Concepts Related to Vision and Image Formation. Physics, 4(4), 1117–1134. |
Thank you for this comment. Indeed, these very recent articles offer a new dimension to this field of research. In the revised manuscript we added the following text:
p. 3, lines 111-116 – In a different direction, a study compared the results of two teaching interventions for concepts and phenomena in optics, including refraction. The first teaching intervention was conducted through a phenomenological theoretical framework while the second through a classical model-based approach. The data of this study showed that both the students' performance on refraction as well as their interest were better in the phenomenological framework than in the model-based approach.
p. 16, lines 613-616 – However, we have to take into account that the approach of this teaching was model-based. In recent years the emergence of a phenomenological perspective might perhaps give new dimensions to the teaching and learning of refraction.
Reference list
Fliegauf, K., Sebald, J., Veith, J. M., Spiecker, H., & Bitzenbauer, P. (2022). Improving Early Optics Instruction Using a Phenomenological Approach: A Field Study. Optics, 3(4), 409–429.
Sebald, J., Fliegauf, K., Veith, J., Spiecker, H., & Bitzenbauer, P. (2022). The World through My Eyes: Fostering Students’ Understanding of Basic Optics Concepts Related to Vision and Image Formation. Physics, 4(4), 1117–1134.
|

Reviewer 3 Report
The paper “Mental representations and cognitive schemata of ninth grade 2 students for the refraction of light” escribes a study conducted on 213 9th grade students about the transition from mental representations to schemata for what concerns the refraction of light. 3 tasks were made where students were asked to predict the path of a light ray. To determine the general structure of students’ schemata about thee Multiple Correspondence Analysis (MCA) method has been applied.
​The paper is interesting and clear. I add some suggestions to improve the contextualization of the work.
1) Please, add some more information on the teaching of optics students received (hours, examples made, experimental parts etc.) The information in the work is in my opinion, not enough.
2) Explain the reasons for the choices made in the three tasks. In particular, the reasons it was decided to draw the line normal to the separation surface, that, in my opinion makes the drawings a mixture of pictorial representation of the experimental situation, and of a interpretative model.
3) Motivate why it was decided to completely neglect the reflected beam of light.
4) Personally I do not appreciate the “confusion” between collimated light beam and light ray (see for instance p.4, where you write: “A thin beam of light (light ray) and a light ray”). The first is an experimental set up, the second a theoretical concept.
Pag. 4 - I think that girls are supposed to be 110 and not 11.
Author Response
We are very grateful to Reviewer 3 for his very constructive and timely comments.
We have tried to work systematically to respond to all of them.
|
Reviewer 3 |
Authors |
|
1. Please, add some more information on the teaching of optics students received (hours, examples made, experimental parts etc.) The information in the work is in my opinion, not enough. |
Thank you for this comment. In the revised manuscript we added the following text
p. 4, lines 172-180 – The sample of the study consisted of students who attended the third grade of secondary school. According to the Greek curriculum physics is taught in this grade for two hours per week. The chapter of optics, which is the last chapter of the Physics textbook, deal mainly with refraction, reflection as well as Convex and Concave Mirrors. These phenomena are taught in a semi-quantitative manner through Snell's law and textbook exercises. Experimental exercises are confined on apparatus such as the Newton disc, also known as the disappearing colour disc, and the dispersive prism. It should be noted that the phenomenon of refraction had already been taught in the sixth grade of primary school in a totally qualitative manner.
|
|
2. Explain the reasons for the choices made in the three tasks. In particular, the reasons it was decided to draw the line normal to the separation surface, that, in my opinion makes the drawings a mixture of pictorial representation of the experimental situation, and of an interpretative model. |
Thank you for your comment which enables us to explain our choice. It is well known that as literature suggests, the simultaneous use of realistic representations and model elements should be avoided in teaching. However, the students that participated in the current study were so far taught refraction through educational materials (textbooks, notes, etc.) that always had the vertical line drawn and therefore we did not want to deviate the virtual environment in which the students were trained. Our choice was reinforced by the fact that we used these tasks for research rather than teaching purposes.
|
|
3. Motivate why it was decided to completely neglect the reflected beam of light. |
The current study is part of a series of related research on the teaching and learning of optics phenomena and concepts. It is a fact that this article deals exclusively with refraction phenomenon. It will be followed however by another study conducted on a different sample of students that focuses on reflection phenomenon.
|
|
4. Personally, I do not appreciate the “confusion” between collimated light beam and light ray (see for instance p.4, where you write: “A thin beam of light (light ray) and a light ray”). The first is an experimental set up, the second a theoretical concept. |
Thank you for this comment. The term ‘light ray’ was wrongly confused with ‘beam’ due to the fact that in Greek language we often use the term ‘beam of rays’. However, as it looks confusing in the English text, we decided to delete it in the revised manuscript.
|
|
5. Pag. 4 - I think that girls are supposed to be 110 and not 11. |
We would like to apologize as due to a typographical error we mistakenly wrote in the text that11 girls participated in the study, whereas in fact there were 111 girls. It goes without saying that we fixed that mistake in the revised manuscript.
|

Round 2
Reviewer 2 Report
Paper is ready for publication! Congratulations!